# A Novel Method for Constructing Spatiotemporal Knowledge Graph for Maritime Ship Activities

Cunxiang Xie [1], Limin Zhang [1] and Zhaogen Zhong [2,*]

[1] Department of Information Fusion, Naval Aviation University, Yantai 264001, China; xiecunxiang1996@163.com (C.X.); iamzlm@163.com (L.Z.)

[2] The School of Aviation Basis, Naval Aviation University, Yantai 264001, China

[*] Correspondence: zhongzhaogen@163.com

**Abstract:** This study focused on the construction of a spatiotemporal knowledge graph for ship activities. First, a ship activity ontology model was proposed to describe the entities and relations of ship activities. Then, maritime event text data were utilized as the ship activity dataset, where entities and relations were extracted to form triplets. Thus, the data layer was populated, completing the construction of the ship activity spatiotemporal knowledge graph. The process of extracting triplets involved initially inputting the text sentences into the Bidirectional Encoder Representations from Transformers (BERT) model for pretraining to obtain vector representations of characters. These representations were then fed into a lattice long short-term memory network (Lattice-LSTM) for further processing. The resulting hidden vectors $h_1$, $h_2$, $\cdots$, $h_n$, were input into the conditional random field (CRF) to perform named entity recognition. The recognized entities were then labeled in the original sentences and input into another BERT-Lattice-LSTM network. The resulting hidden vectors $h'_1$, $h'_2$, $\cdots$, $h'_n$ were fed into a relation classifier, which output the relation between the two labeled entities, completing the extraction of entity–relation triplets. In experiments, the proposed method achieved triplet extraction performance exceeding 90% for three different evaluation metrics: Precision, Recall, and F1-measure.

**Keywords:** ship activities; spatiotemporal knowledge graph; ontology model; BERT; Lattice-LSTM

## 1. Introduction

With global economic integration, the number of ships has been continuously increasing in recent years, which has led to an increase in the frequency of maritime ship activities and accidents [1]. To ensure the safe and efficient operation of maritime traffic, maritime authorities and coastal defense departments of various countries have jointly established the global automatic identification system (AIS) for maritime traffic monitoring, aiming to enhance users' comprehensive situational awareness of global maritime traffic [2]. Researchers have applied Big Data analytics techniques to ship trajectory analysis, facilitating the intelligent development of maritime traffic monitoring and management [3]. Traditional methods mainly rely on ship positioning data to mine routine ship activities, without incorporating in-depth analysis of sudden events and accidents using other multi-source maritime event data; thus, they lack in-depth knowledge mining. Therefore, there is demand for leveraging cutting-edge technologies such as artificial intelligence to strengthen maritime traffic monitoring and management and to mine data of ship trajectories and multi-source maritime events [4].

The main focus of previous research was mining semantic information from ship trajectory data for ship activity behavior analysis [5]. Scholars have explored ship activity patterns via event analysis using ship event data [6]. However, the models that describe ship events take events as the basic unit and cannot effectively represent a single voyage activity process and the basic behavior of a ship. To analyze the causes of ship events

in-depth and express ship activities more comprehensively, it is often necessary to combine the entire set of events during a ship's navigation process and the ship's behavioral information before and after the events. If the method of event extraction is employed to parse the textual data describing the events and analyzed together with trajectory data, a more complete representation of ship activities can be obtained. Constructing a knowledge graph [7] is an effective way to integrate multi-source data. A knowledge graph is a knowledge repository that represents entities (concepts, people, and things) in the objective world and their relations in the form of a graph. Essentially, a knowledge graph is a large-scale semantic network comprising an ontology layer and a data layer. The ontology layer describes conceptual entities and the relations between them, while the data layer stores real-world entities and relations. In addition to semantic knowledge of entities, a spatiotemporal knowledge graph [8] focuses on representing temporal and spatial relations. The present study primarily focuses on the construction of a spatiotemporal knowledge graph for maritime ship activities. The maritime ship activity spatiotemporal knowledge graph is a knowledge repository that represents the temporal and spatial maritime activities of ships and the relations between them in the form of a graph, with maritime ship activity entities as nodes and the relations between them as edges. The construction of the maritime ship activity knowledge graph relies on the ontology layer to describe conceptual entities and their relations. Multiple sources of ship activity data are then populated into the data layer, completing the construction of the maritime ship activity spatiotemporal knowledge graph.

Once the ontology layer is constructed, the data layer needs to be populated through two techniques: named entity recognition and relation extraction. Named entity recognition methods can be divided into two groups: (1) rule-based methods that rely on feature engineering and domain knowledge and (2) traditional machine-learning methods. Rule-based methods were commonly used in early Chinese-language named entity recognition. This approach requires manual rule construction and has a strong dependence on domain knowledge, making rule creation and modification time-consuming and labor-intensive. With the rise of machine-learning methods, the manual rule construction process in rule-based methods has been incorporated into post-processing of named entity recognition models based on machine-learning methods. Machine-learning methods mainly include support vector machines [9], hidden Markov models [10], and conditional random fields (CRFs) [11]. These methods still require additional features. For English named entity recognition tasks, neural networks have become the mainstream approach—particularly convolutional neural network–conditional random fields (CNN-CRF) [12–15] and bidirectional long short-term memory–conditional random fields (BiLSTM-CRF) [16–18].

Relation extraction involves automatically identifying the types of semantic relations between entities. Typically, recurrent neural network (RNN) architectures are used to model the complex interactions and contextual information among entities and their mentions in the document, capturing entity information and generating entity representations. Finally, according to these representations, the model predicts the entity relation types. Currently, long short-term memory (LSTM) and bidirectional long short-term memory (Bi-LSTM) networks based on the RNN architecture can effectively capture long-distance interactions between entities in the document. With the introduction of attention mechanisms, the model can focus on the information related to the target entities in the sentence, achieving more efficient entity relation classification [19–21].

Existing named entity recognition and relation extraction techniques have achieved excellent performance on English documents. However, this study focuses on Chinese maritime intelligence information, which lacks explicit word boundary information compared with English. Nevertheless, word boundary information and semantic information are crucial for Chinese named entity recognition and relation extraction tasks. To address this issue, we propose using the Lattice-LSTM model [22,23] to represent dictionary words in the sentence, integrating the implicit lexical features into the character-based LSTM model. The sentence is matched with an automatically acquired dictionary to construct a

word-based Lattice-LSTM model, which is trained using Bidirectional Encoder Representations from Transformers (BERT) on a large-scale Chinese text after segmentation. The trained output dictionary can help solve deep-level named entity recognition and relation extraction problems in the context.

This paper presents a method for constructing a spatiotemporal knowledge graph specifically for ship navigation activities. Our contributions are summarized as follows:

(1) An ontology model based on the SEM is proposed for ship navigation activities, which represents the hierarchical relations among concepts related to ship activities. It includes six core entity concepts: process, event, actor, place, time, and action. Additionally, according to the characteristics of ship activities, entity relations are defined, including hasEvent, hasActor, hasPlace, hasTime, hasAction, cause, and followed.

(2) A BERT-Lattice-LSTM network model was developed. Initially, the BERT model was utilized to pretrain the textual data of maritime activities, generating vector representations for characters. Subsequently, the Lattice-LSTM model was presented to represent dictionary words in the sentences, integrating implicit lexical features into the character-based LSTM model.

(3) A method was developed for extracting ship activity triplets. Specifically, two BERT-Lattice-LSTM network models were established: one for named entity recognition and the other for relation classification. The maritime activity intelligence text was fed into the named entity recognition model based on the BERT-Lattice-LSTM network. The hidden-layer output $h_1$, $h_2$, $\cdots$, $h_n$ was then input into a CRF model to achieve named entity recognition. The recognized named entities were marked in the maritime activity intelligence text and fed into the relation extraction model based on the BERT-Lattice-LSTM network. The hidden-layer output $h'_1$, $h'_2$, $\cdots$, $h'_n$ was input into a relation classifier (RC), which output the type of relation between the two named entities; thus, the ship activity triplets were obtained.

(4) Experiments were designed to compare the proposed model with four other models: LSTM-CRF-RC, BiLSTM-CRF-RC, BERT-LSTM-CRF-RC, and BERT-BiLSTM-CRF-RC. The proposed model achieved superior performance in named entity recognition, relation extraction, and triplet extraction. The results confirmed the effectiveness of the proposed method.

## 2. Research Methods and Materials

### 2.1. Design of Ontology Rules

The ontology layer is constructed using a concept-based simple event model (SEM), while the data layer is populated with knowledge triplets obtained through named entity recognition and relation extraction techniques. Ship activity events can modeled using general event models. In current academic research, concept-based event models [24–27], logic-based hierarchical event models [28–31], and sextuplet-based event models [32–35] are used. The modeling of ship activities in this study belongs to the category of domain-specific ontology modeling, where concept-based event models such as ABC ontology [24] model, SEM [25], EO [26] model, and CIDOC-CRM [27] model are primarily used.

The ABC ontology model focuses on modeling event concepts and expresses the relations between concepts such as events, scenarios, actions, and objects to describe event content. It classifies entities into abstraction, actuality, and temporality classes and considers the time, place, and agent information of events. In particular, the actuality class describes the objective existence in the real world, while the temporality class describes entities with temporal existence. The situation class represents a contextual environment and expresses the temporal dependency of actuality entities. The event class represents the transition between situations and is related to the situation class through the preceding and following attributes. It is also associated with the action and agent classes.

The SEM ontology model proposed by Hage [25] represents and infers events through the definition of classes, properties, and constraints. Its core classes include event, actor, place, and time, with the aim of enhancing the model's generality, and other events can be added based on this foundation. The properties are divided into three types: event proper-

ties, type properties, and other properties (such as sub-properties, e.g., "according To" and "has Time Stamp"). Each core class has a type property for easy querying. "sem:according To" associates "sem:View" with "sem:Authority" to express different viewpoints and opinions. "sem:has Time Stamp" has seven sub-properties: one for expressing a single-valued time, i.e., "sem:has Time Stamp"; two for representing time intervals, i.e., "sem:has Begin Time Stamp" and "sem:has End Time Stamp"; and four for representing uncertain time intervals, i.e., "sem:has Earliest Begin Time Stamp", "sem:has Latest Begin Time Stamp", "sem:has Earliest End Time Stamp", and "sem:has Latest End Time Stamp".

The EO model primarily consists of four classes (event and three implicit classes: agent, factor, and product) and seventeen attribute groups. It defines the minimum number of events and relies on an external vocabulary to refine the expressed knowledge. Similar to the SEM, the EO model adopts a modular design, which enhances its flexibility. However, it lacks explicit actor and place classes. Meanwhile, CIDOC-CRM is a concept-based, large-scale ontology with no formal restrictions. It comprises 140 classes and 144 attributes, and a subset of these can be used to represent events.

The SEM generalizes the CIDOC-CRM model and introduces the concept of view. Additionally, it provides lightweight descriptive elements for events; however, it avoids introducing strongly defined semantics that can lead to inconsistencies. Moreover, it leverages types, constraints, and authority to facilitate the integration of external data. Hage used the SEM to model and identify ship events according to ship trajectory data, allowing the transformation of ship trajectory data into semantic information about ship events. However, solely the SEM is used for event extraction, only regular ship events can be described, and the relation between ship activity processes and behaviors cannot be captured.

The ship activity components in this study include processes, events, and behavioral elements. Therefore, the aforementioned modeling methods are not fully applicable to modeling the ship activities in this study. It is possible to extend the SEM according to the actual composition of ship activity components. Because the SEM expresses minimal events and is easily expandable, it has been applied to ship trajectory data, confirming its feasibility.

The ontology is an essential component of a knowledge graph and can formally represent the hierarchical relations among concepts related to ship activities. The SEM [25] is a domain-independent event representation model that can be applied to model events in different domains. It describes events using core concepts, class systems, and attribute constraints. It comprehensively utilizes four concepts—time, place, object, and event—to describe the components of an event. By setting class systems corresponding to core concepts, the class information of event elements can be described using specific instances without changing the pattern layer. Attribute constraints are used to describe properties in the knowledge graph. By adding information to existing attributes, they can be constrained or expanded with regard to their descriptions. In this study, the class system and attribute constraint rules of the SEM are utilized, and the concept system, entity classes, and entity relations are supplemented using the Web Ontology Language (OWL). A ship activity model (SAM) is proposed, which includes processes, events, and actions related to ship activities.

The SAM model consists of six core entity concepts:

(1) Process, which represents the sea voyage process of a ship from one port to another, including transportation, fishing, cruising, escorting, etc.

(2) Event, which represents the reasons for changes in the maritime status of a ship, including natural disasters, maritime accidents, and other incidents.

(3) Actor, which represents the subject participating in the event, i.e., the ship.

(4) Place, which represents entities with spatial locations, such as specific place names or coordinates.

(5) Time, which represents entities with time characteristics, such as a specific point in time or a time interval.

(6) Action, which represents the fundamental actions of ship activities, such as anchoring, movement (uniform, accelerating, and decelerating), and mooring.

According to the characteristics of ship activities, the entity relations can be defined as shown in Table 1.

**Table 1.** Semantic association between ship activities.

| Relation | Subject(s) | Object(s) |
| --- | --- | --- |
| sam: hasEvent | Process | Event |
| sam: has Actor | Process<br>Event<br>Action | Actor<br>Actor<br>Actor |
| sam: hasPlace | Process<br>Event<br>Action | Place<br>Place<br>Place |
| sam: hasTime | Process<br>Event<br>Action | Time<br>Time<br>Time |
| sam: has Action | Event | Action |
| sam: cause | Event | Event |
| sam: followed | Action | Action |

The core concepts of the SAM model and their relations are shown in Figure 1.

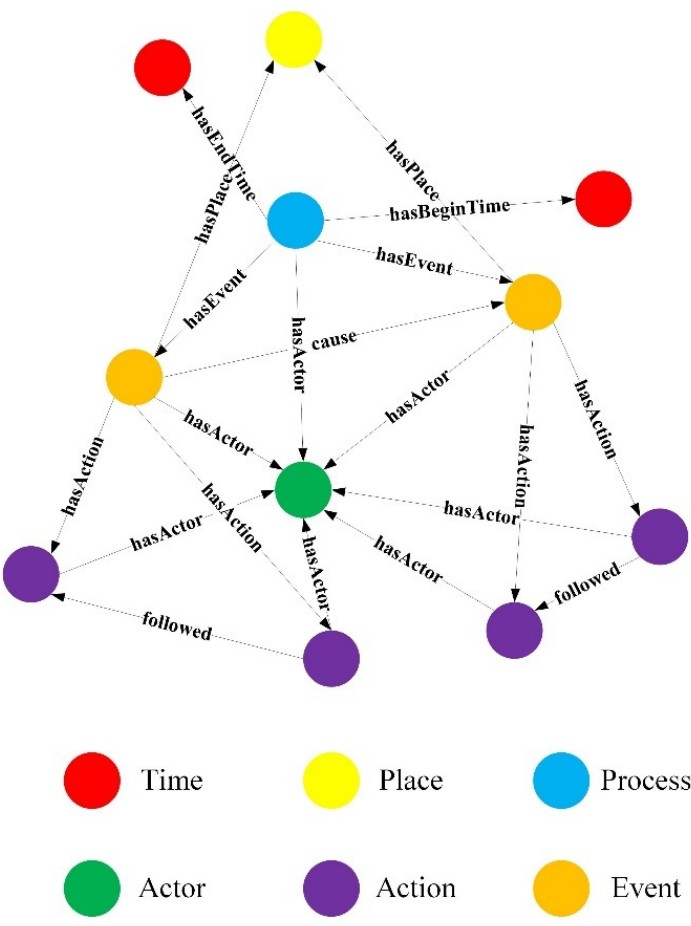

**Figure 1.** Ship activity knowledge graph.

### 2.2. BERT Model

BERT is a self-supervised deep language model that trains text using a multilayer bidirectional transformer encoding structure with a masking mechanism [36]. The transformer encoder is composed of a self-attention mechanism and a feedforward neural network, which eliminates the recurrent structure and allows parallel computation. In contrast to previous models such as RNNs and LSTMs, BERT allows concurrent execution and extraction of word relation features in a sentence. It can extract relation features at multiple levels, providing a more comprehensive reflection of sentence semantics. Additionally, in contrast to previous pretraining models, BERT can capture word meanings according to sentence context, avoiding ambiguity. Furthermore, BERT can extract word meanings in both directions, resulting in richer and more implicit features. The overall structure of the BERT model is illustrated in Figure 2.

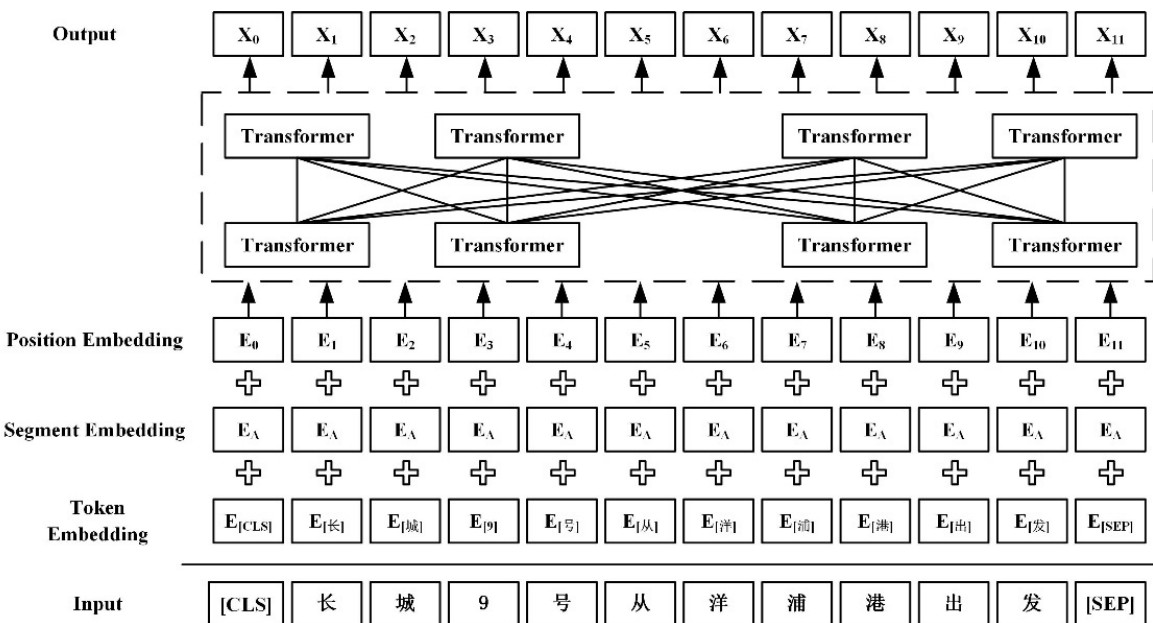

**Figure 2.** The model architecture for BERT pre-training.

In BERT, the input text is first transformed into semantic vectors. This process includes token embedding, segment embedding, and position embedding, which are combined. Token embedding converts the input text sequence into fixed-dimensional vectors, segment embedding incorporates information from different sentences, and position embedding encodes the sequential order of the input text sequence. These embeddings are then passed to multiple transformer encoders for training, resulting in trained word vectors. The most important structure in BERT is the transformer encoder, which includes key operations such as multi-head attention, self-attention, residual connections, layer normalization, and linear transformations. Through these operations, the transformer encoder transforms the semantic vectors of individual words in the input text into enhanced semantic vectors of the same length. With multiple layers of transformer encoders, BERT achieves the training of semantic vectors for each word in the text.

For BERT, the crucial component is the transformer structure. The transformer is a deep network based on the self-attention mechanism, which is the key part. It adjusts the weight coefficient matrix of word associations within the one sentence to obtain word representations. The corresponding formula is

$$\text{Attention}(\boldsymbol{Q}, \boldsymbol{K}, \boldsymbol{V}) = \text{Softmax}\left(\frac{\boldsymbol{Q}\boldsymbol{K}^{T}}{\sqrt{d_k}}\right)\boldsymbol{V} \tag{1}$$

where $Q$, $K$, and $V$ represent the matrix of word vectors, $QK^T$ represents the dot-product matrix of $Q$ and $K^T$ reflecting the degree of association of each word with another word, $\sqrt{d_k}$ is the scale factor, and $d_k$ represents the dimensionality of the word vectors.

Building upon this, multiple self-attention layers are concatenated through a multi-head structure to achieve a more interpretable multi-head attention mechanism. The corresponding formulas are as follows:

$$\text{MultiHead}(\boldsymbol{Q}, \boldsymbol{K}, \boldsymbol{V}) = \left[\text{head}^1; \text{head}^2; \cdots ; \text{head}^n\right] \boldsymbol{W} \tag{2}$$

$$\text{head}^i = \text{Attention}n\left(\boldsymbol{Q}\boldsymbol{W}_Q^i, \boldsymbol{K}\boldsymbol{W}_K^i, \boldsymbol{V}\boldsymbol{W}_V^i\right) \tag{3}$$

where $\boldsymbol{W}$ represents the weight matrix; $\boldsymbol{W}_Q$, $\boldsymbol{W}_K$, and $\boldsymbol{W}_V$ represent the weight matrices of $\boldsymbol{Q}$, $\boldsymbol{K}$, and $\boldsymbol{V}$, respectively.

The advantage of the BERT model lies in its inclusion of two tasks [37]: masked language modeling (MLM) and next sentence prediction (NSP). The basic idea of the MLM is to randomly mask words, with most of the masked words being replaced with "[MASK]", some being randomly replaced, and the remainder kept unchanged. Through joint training, the model can infer the masked words according to the context, addressing the issue of word ambiguity. In contrast, NSP provides an intuitive understanding of the logical relation between preceding and subsequent sentences. The combination of these two tasks enhances the semantic representation of the model.

### 2.3. Lattice-LSTM Structure

RNNs are commonly used for processing sequential data, such as textual data. They allow computers to understand sequential data from a holistic perspective. However, owing to the issue of vanishing gradients, RNNs fail to capture long-range contextual features. LSTMs were introduced to address this problem. They employ a gated strategy to solve the vanishing-gradient problem and other issues during backpropagation. This approach is commonly used in many natural language processing tasks. With regard to structural composition, LSTM is similar to CNNs, with the difference lying in the use of complex network graphs during each recurrent computation in LSTM. The LSTM network structure consists mainly of four gate units that interact with each other in a special way. The computational process is expressed by Equations (4)–(9) [38]:

$$f_j^c = \sigma\left(W_f^c x_j^c + U_f^c h_{j-1}^c + b_f^c\right) \tag{4}$$

$$o_j^c = \sigma\left(W_o^c x_j^c + U_o^c h_{j-1}^c + b_o^c\right) \tag{5}$$

$$i_j^c = \sigma\left(W_i^c x_j^c + U_i^c h_{j-1}^c + b_i^c\right) \tag{6}$$

$$\widetilde{c}_j^c = \tanh\left(W_{\widetilde{c}}^c x_j^c + U_{\widetilde{c}}^c h_{j-1}^c + b_{\widetilde{c}}^c\right) \tag{7}$$

$$c_j^c = f_j^c \odot c_{j-1}^c + i_j^c \odot \widetilde{c}_j^c \tag{8}$$

$$h_j^c = o_j^c \odot \tanh\left(c_j^c\right) \tag{9}$$

where $f_j^c$, $o_j^c$, and $i_j^c$ denote the forget gate, output gate, and input gate, respectively; $W_f^c$, $W_o^c$, $W_i^c$, $W_{\widetilde{c}}^c$, $U_f$, $U_o$, $U_i$, $U_{\widetilde{c}}$, $b_f$, $b_o$, $b_i$, and $b_{\widetilde{c}}$ are the model parameters; $\widetilde{c}_j^c$ represents the new candidate value of the cell state; $c_j^c$ represents the new cell information obtained through the input gate; $h_j^c$ represents the output of the LSTM model; and $\sigma(\cdot)$ and $\tanh(\cdot)$ are different

neuron activation functions. This gating approach allows effective selection and extraction of associated information from memory units, addressing the fatal flaw of RNNs.

One limitation of the character-based LSTM model in handling named entity recognition and relation extraction tasks is that the representation of character and character position information is not effectively utilized. Therefore, external sources of information are needed to perform named entity recognition and relation extraction. To address this issue, this paper proposes using the Lattice-LSTM model to represent dictionary words in sentences and integrate the implicit lexical features into the character-based LSTM model. Here, an automatically obtained dictionary is matched with the sentence to construct the word-based Lattice-LSTM model, which is derived from a large-scale Chinese text that has been segmented and trained using BERT. The trained output dictionary can be used to solve deep named entity recognition and relation extraction problems in context.

Let $w_{b,e}^d$ be a word in the dictionary, with $b$ being the start position of the word, $e$ being the end position of the word, and $x_{b,e}^d$ being the word vector:

$$x_{b,e}^d = e^w\left(w_{b,e}^d\right) \tag{10}$$

Here, $e^w(\cdot)$ is the word vector mapping table. In addition, the state of the memory unit of $x_{b,e}^d$ is recorded together with $c_{b,e}^d$:

$$f_{b,e}^w = \sigma\left(W_f^w x_{b,e}^w + U_f^w h_b^c + b_f^w\right) \tag{11}$$

$$i_{b,e}^w = \sigma\left(W_i^w x_{b,e}^w + U_i^w h_b^c + b_i^w\right) \tag{12}$$

$$\widetilde{c}_{b,e}^w = \tanh\left(W_{\widetilde{c}}^w x_{b,e}^w + U_{\widetilde{c}}^w h_b^c + b_{\widetilde{c}}^w\right) \tag{13}$$

$$c_{b,e}^w = f_{b,e}^w \odot c_b^c + i_{b,e}^w \odot \widetilde{c}_{b,e}^w \tag{14}$$

where $i_{b,e}^w$ and $f_{b,e}^w$ denote the input gate and forget gate, respectively. The Lattice-LSTM model extracts information from characters, similar to the LSTM model, but for word information extraction, the LSTM model is redesigned by incorporating an external dictionary to enhance its ability to capture word information. This model integrates word sequence information and an additional gate $i_{b,e}^c$, which is used to control the information flow:

$$i_{b,e}^c = \sigma\left(W_i^c x_b^c + U_i^c c_{b,e}^w + b_i^c\right) \tag{15}$$

All the $c_{b,e}^w$ and $\widetilde{c}_j^c$ values are used to calculate $c_j^c$:

$$c_j^c = \sum_{b\in\{b'|w_{b',j}^d\in D\}} \alpha_{b,j}^c \odot c_{b,j}^w + \alpha_j^c \odot \widetilde{c}_j^c \tag{16}$$

where $D$ represents the dictionary set. $\alpha_{b,j}^c$ and $\alpha_j^c$ can be calculated as follows:

$$\alpha_{b,j}^c = \frac{\exp\left(i_{b,e}^c\right)}{\exp(i_e^c) + \sum_{b'\in\{b''|w_{b'',e}\in D\}}\exp\left(i_{b',e}^c\right)} \tag{17}$$

$$\alpha_e^c = \frac{\exp(i_e^c)}{\exp(i_e^c) + \sum_{b'\in\{b''|w_{b'',e}\in D\}}\exp\left(i_{b',e}^c\right)} \tag{18}$$

By substituting the $c_j^c$ calculated via Equation (16) into Equation (9), $h_j^c$ is obtained.

### 2.4. Named Entity Identification and Relation Classification

For named entity recognition, the final label prediction is typically given by the network output layer in its compositional structure, which normalizes the non-standardized calculation values from the hidden-layer output. In simple terms, it transforms the model's scores for different labels into probabilities and provides the final classification prediction. However, the probability calculation for each label result is independent, and the local labels and contextual information are not considered in the normalization function. Thus, using a normalization function is not the most accurate strategy. To address this issue, we propose the CRF model, which considers the relevance of neighboring labels and achieves more accurate labeling of sentence-level information by incorporating relevant label data. Thus, the output $h_1, h_2, \cdots, h_n$ of the Lattice-LSTM model is fed into the CRF model to calculate the probability value of the label sequence $y_e = l_1, l_2, \cdots, l_n$ [23]:

$$P(y_e|s) = \frac{\exp\left(\sum_i \left(W_{CRF}^{l_i} h_i + b_{CRF}^{(l_{i-1}, l_i)}\right)\right)}{\sum_{y'_e} \exp\left(\sum_i \left(W_{CRF}^{l'_i} h_i + b_{CRF}^{(l'_{i-1}, l'_i)}\right)\right)} \tag{19}$$

where $y'_e$ represents an arbitrary sequence of labels, $W_{CRF}^{l_i}$ represents the model parameters specific to $l_i$, $b_{CRF}^{(l_{i-1}, l_i)}$ and represents the biases specific to $l_{i-1}$ and $l_i$.

The Viterbi algorithm is called to find the highest-scoring label sequence from the input sequence. Given a set of manually labeled training data $\{(s_i, y_{ei})\}|_{i=1}^N$, the model is trained using the $L_2$-regularized sentence-level log-likelihood loss:

$$L_e = \sum_{i=1}^N \log(P(y_{ei}|s_i)) + \frac{\lambda}{2}\|\Theta\|^2 \tag{20}$$

where $\lambda$ denotes $L_2$ regularization, and $\Theta$ represents the set of the parameters to be trained in the model.

For the output $\boldsymbol{h} = (h_1, h_2, \cdots, h_n) \in R^{d_h \times n}$ of the Lattice-LSTM model, where $d_h$ represents the dimensionality of the output vector $h_j$ of the model, we first merge them into a sentence-level eigenvector $\boldsymbol{h}^* \in R^{d_h}$ using the character-level attention mechanism and then feed $\boldsymbol{h}^*$ into the RC to calculate the confidence of each class of relations. The sentence-level eigenvector $\boldsymbol{h}^*$ can be calculated as follows [39]:

$$\boldsymbol{H} = \tanh(\boldsymbol{h}) \tag{21}$$

$$\boldsymbol{\alpha} = \text{softmax}\left(\boldsymbol{\omega}^T \boldsymbol{H}\right) \tag{22}$$

$$\boldsymbol{h}^* = \boldsymbol{h}\boldsymbol{\alpha}^T \tag{23}$$

The conditional probability of the relation class $y_r$ corresponding to a given sentence $S$ can be calculated as follows:

$$P(y_r|S) = \text{softmax}(\boldsymbol{W}\boldsymbol{h}^* + \boldsymbol{b}) \tag{24}$$

where $\boldsymbol{W} \in R^{Y \times d_h}$ represents the transformation matrix, $\boldsymbol{b} \in R^Y$ represents the bias vector, and $Y$ represents the total number of relation classes.

Given a manually labeled training dataset $\{(s_i, y_{ri})\}|_{i=1}^N$, the model can be trained using the sentence-level log-likelihood loss:

$$L_r = \sum_{i=1}^N \log(P(y_{ri}|s_i)) \tag{25}$$

### 2.5. Triplet Extractor

We constructed two BERT-Lattice-LSTM network models: one for named entity recognition and the other for relation classification. First, the training for named entity recognition was conducted. In this step, part-of-speech tagging was performed on the model input. The Chinese sentence was annotated using the BIOSE tagging scheme, where each character is labeled as follows: B (Begin) represents the start of a named entity, I (Inside) represents the inside of a named entity, O (Other) represents non-entity characters, S (Single) represents a single character entity, and E (End) represents the end of a named entity. An example is "长(B-Actor)城(I-Actor)9(I-Actor)号(E-Actor)从(O)洋(B-Place)浦(I-Place)港(E-Place)出(B-Event)发(E-Event)". Finally, model training was conducted using Equation (20).

Taking the Chinese sentence "长城9号从洋浦港出发" (literally, Great Wall 9 departs from Yangpu Port) as an example, Figure 3 shows the named entity recognition model based on the BERT-Lattice-LSTM network.

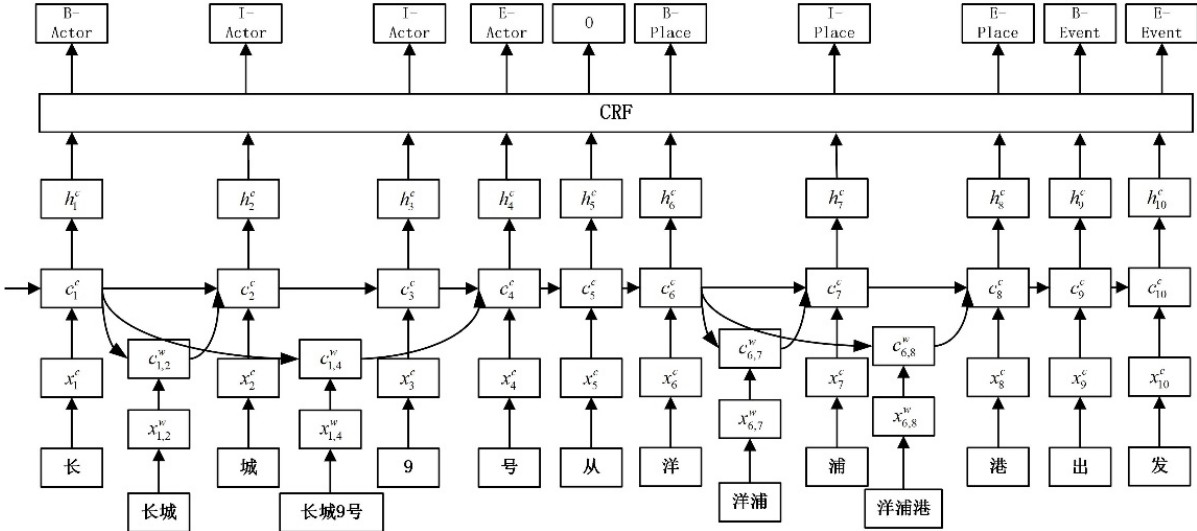

**Figure 3.** Named entity recognition model based on the BERT-Lattice-LSTM network.

Named entities obtained through named entity recognition in the Chinese sentence are labeled within the sentence and fed into the relation classification model based on the BERT-Lattice-LSTM network. The output represents the relation between the two labeled named entities.

Similarly, taking the Chinese sentence "长城9号从洋浦港出发" (literally, Great Wall 9 departs from Yangpu Port) as an example, Figure 4 illustrates the relation classification model based on the BERT-Lattice-LSTM network.

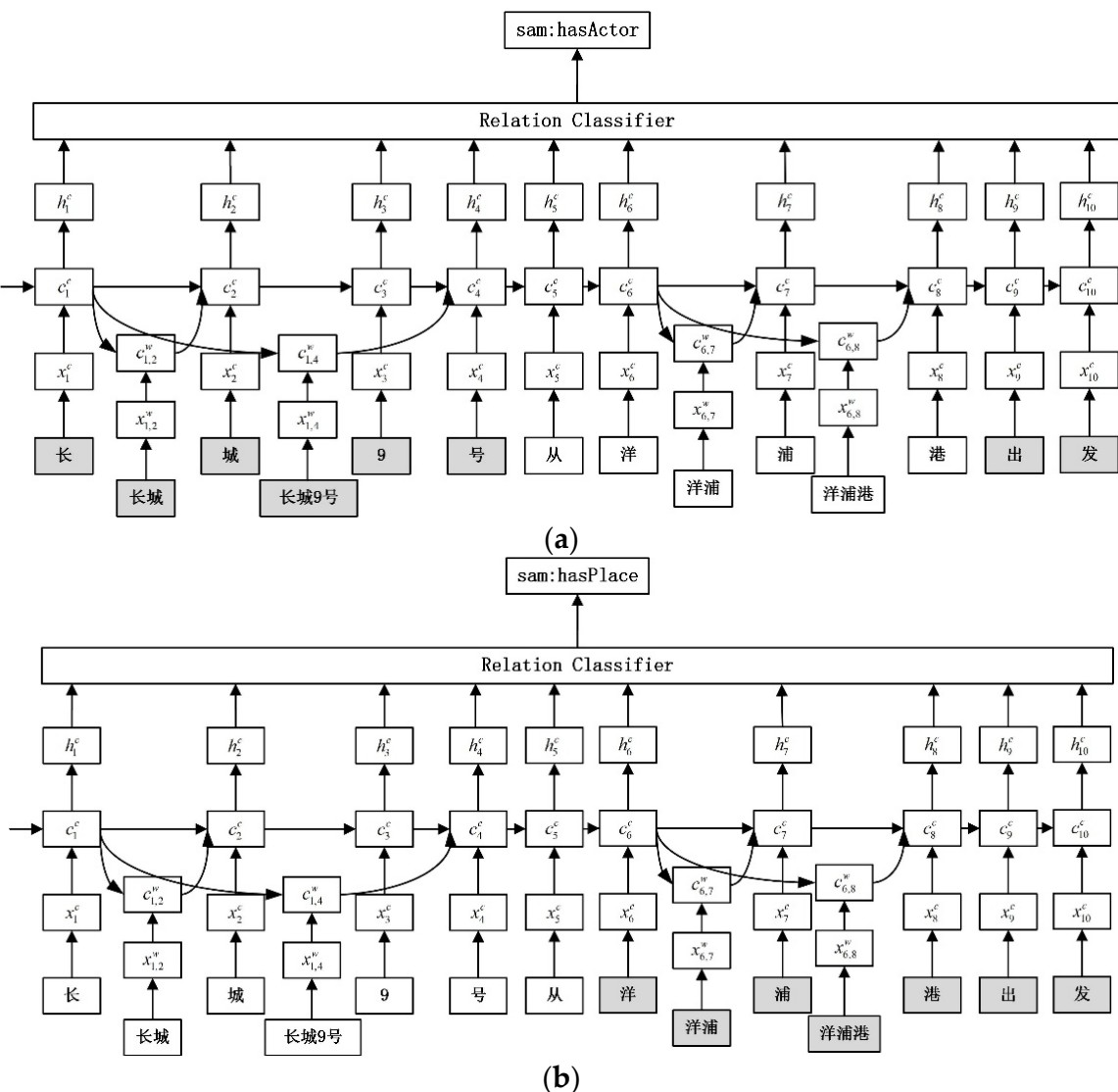

**Figure 4.** Named entity recognition model based on the BERT-Lattice-LSTM network. (**a**) Labeled entities are "长城9号" (Great Wall 9) and "出发" (departs). (**b**) Labeled entities are "洋浦港" (Yangpu Port) and "出发" (departs).

## 3. Experiment

### 3.1. Datasets and Evaluation Metrics

In the experiment, ship event text data from Shipxy (www.shipxy.com) were used as the ship activity dataset. A total of 2548 sentences were selected and labeled with entity classes and relation classes for named entity recognition and relation extraction. The resulting dataset was divided into training, validation, and test sets at a ratio of 7:2:1.

The evaluation metrics used for named entity recognition and relation extraction were the Precision, Recall, and F1-measure. In the case of binary classification, the true classifications of the test dataset were compared with the model's predicted classifications. The four metrics are shown in Figure 5, with the total sample count given as $TP + FP + FN + TN$. The comparison results were represented using a confusion matrix.

Precision is the ratio of the number of correctly recognized named entities (relation classes) to the total number of recognized named entities (relation classes). It is calculated as follows:

$$Precision = \frac{TP}{TP + FP} \tag{26}$$

Recall is the ratio of the number of correctly recognized named entities (relation classes) to the total number of named entities (relation classes) in the dataset. It is calculated as follows:

$$Recall = \frac{TP}{TP + FN} \tag{27}$$

where $A_N$ represents the number of aligned entity pairs in the dataset. A higher recall indicates better model performance.

The F1-measure reflects both the Precision and Recall. It is calculated as follows:

$$F1 = \frac{2 \times Recall \times Prcision}{Recall + Prcision} \tag{28}$$

The F1-measure combines the results of Precision and Recall. A higher F1-measure indicates better overall model performance.

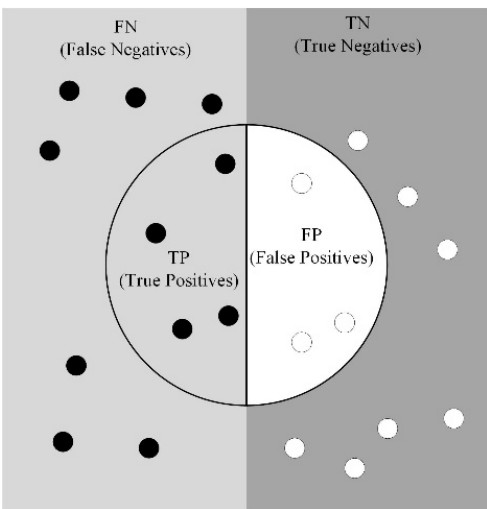

**Figure 5.** Comparison of four prediction metrics.

### 3.2. Experimental Setup

The experiments were conducted using the PyTorch 1.9.0 framework, which is widely used by researchers for implementing various machine-learning algorithms. The model construction and training were implemented using Python. The dimensions of the character vectors and word vectors in this study were set as 768. The Adam optimizer was used, and the learning rate during training was set as 0.01. To prevent exploding gradients during training, the gradient clipping technique was employed, with a parameter value of 5. The dropout technique with a value of 0.5 was used to prevent overfitting.

### 3.3. Named Entity Recognition Performance Validation of Proposed Model

The performance of the proposed named entity recognition model was experimentally evaluated in comparison with the LSTM-CRF, BiLSTM-CRF, BERT-LSTM-CRF, and BERT-BiLSTM-CRF models. The test results for the named entity recognition performance are presented in Table 2.

**Table 2.** Test results for the named entity recognition performance (%).

| Model | Precision | Recall | F1-Measure |
|---|---|---|---|
| LSTM-CRF | 78.15 | 77.07 | 77.61 |
| BiLSTM-CRF | 82.15 | 79.66 | 80.89 |
| BERT-LSTM-CRF | 90.68 | 88.64 | 89.65 |
| BERT-BiLSTM-CRF | 93.38 | 91.55 | 92.46 |
| BERT-Lattice-LSTM-CRF | **96.04** | **95.30** | **95.67** |

As shown in Table 2, the proposed named entity recognition model achieved the highest Precision, Recall, and F1-measure values, indicating its superior performance. Compared with the LSTM-CRF and BiLSTM-CRF models, the BERT-LSTM-CRF model exhibited improvements of 12.53% and 8.53% in Precision, 11.57% and 8.98% in Recall, and 12.04% and 8.76% in F1-measure, respectively. The BERT-BiLSTM-CRF model exhibited improvements of 15.23% and 11.23% in Precision, 14.48% and 11.89% in Recall, and 14.85% and 11.57% in F1-measure, respectively. The BERT-Lattice-LSTM-CRF model exhibited improvements of 17.89% and 13.89% in Precision, 18.23% and 15.64% in Recall, and 18.06% and 14.78% in F1-measure, respectively. These results indicate that utilizing BERT-based pretraining models can enhance the named entity recognition performance. Moreover, compared with the BERT-LSTM-CRF and BERT-BiLSTM-CRF models, the BERT-Lattice-LSTM-CRF model exhibited improvements of 5.36% and 2.66% in Precision, 6.66% and 3.75% in Recall, and 9.02% and 3.21% in F1-measure, respectively. This suggests that incorporating the Lattice-LSTM model can fuse implicit lexical features into the character-based LSTM model and thereby improve the performance of named entity recognition. The experimental results confirm the effectiveness of BERT and Lattice-LSTM models for enhancing the named entity recognition performance.

The data of Table 2 are displayed as a bar chart in Figure 6. As shown, the proposed model outperformed the LSTM-CRF, BiLSTM-CRF, BERT-LSTM-CRF, and BERT-BiLSTM-CRF models with regard to the three evaluation metrics: Precision, Recall, and F1-measure. This confirms the superior performance of the proposed model for named entity recognition.

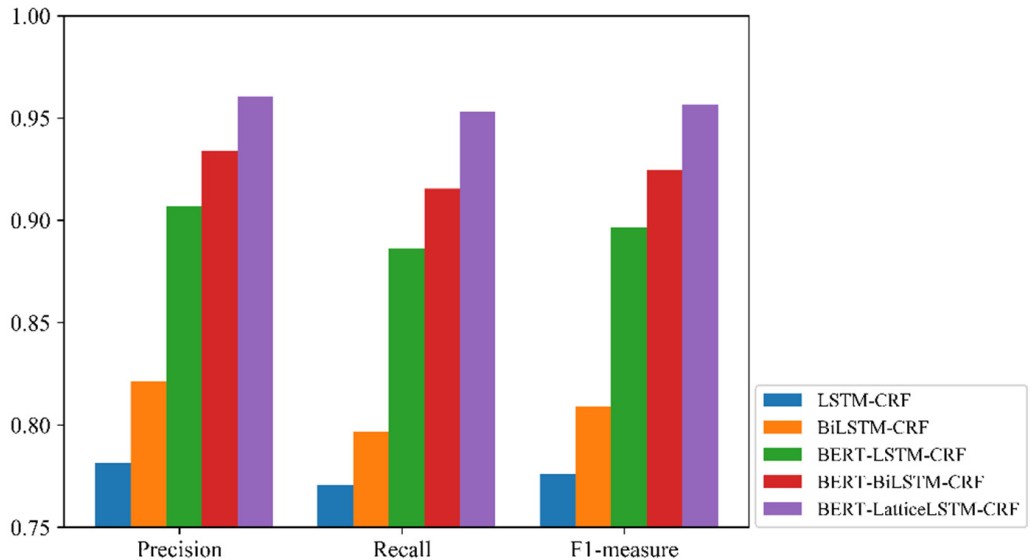

**Figure 6.** Test results for the named entity recognition performance (%).

Furthermore, we conducted tests on six entity types: process, event, actor, place, time, and action, to evaluate the F1-measures of the LSTM-CRF, BiLSTM-CRF, BERT-LSTM-CRF, BERT-BiLSTM-CRF, and BERT-Lattice-LSTM-CRF models in recognizing different entity classes. The test results are presented in Table 3.

As shown in Table 3, compared with the other models, the proposed model achieved F1-measure improvements ranging from 1.05% to 18.32% for the process entity, from 5.62% to 16.77% for the event entity, from 0.76% to 14.84% for the actor entity, from 5.55% to 21.15% for the place entity, from 5.37% to 21.15% for the time entity, and from 0.91% to 18.31% for the action entity. The proposed model achieved the highest F1-measure values for different entity types, further highlighting its superior named entity recognition performance.

The data of Table 3 are displayed as a bar chart in Figure 7. As shown, the proposed model outperformed the LSTM-CRF, BiLSTM-CRF, BERT-LSTM-CRF, and BERT-BiLSTM-CRF models with regard to the F1-measure for all six entity types: process, event, actor,

place, time, and action. This confirms the superior performance of the proposed model for named entity recognition.

**Table 3.** Test results for the named entity recognition performance for different entity types (%).

| Model | Process | Event | Actor | Place | Time | Action |
|---|---|---|---|---|---|---|
| LSTM-CRF | 76.84 | 79.21 | 80.22 | 75.66 | 77.24 | 76.49 |
| BiLSTM-CRF | 81.19 | 79.38 | 82.16 | 82.33 | 79.14 | 81.14 |
| BERT-LSTM-CRF | 88.60 | 87.98 | 91.33 | 92.18 | 88.54 | 89.27 |
| BERT-BiLSTM-CRF | 94.11 | 90.36 | 94.30 | 91.26 | 90.84 | 93.89 |
| BERT-Lattice-LSTM-CRF | **95.16** | **95.98** | **95.06** | **96.81** | **96.21** | **94.80** |

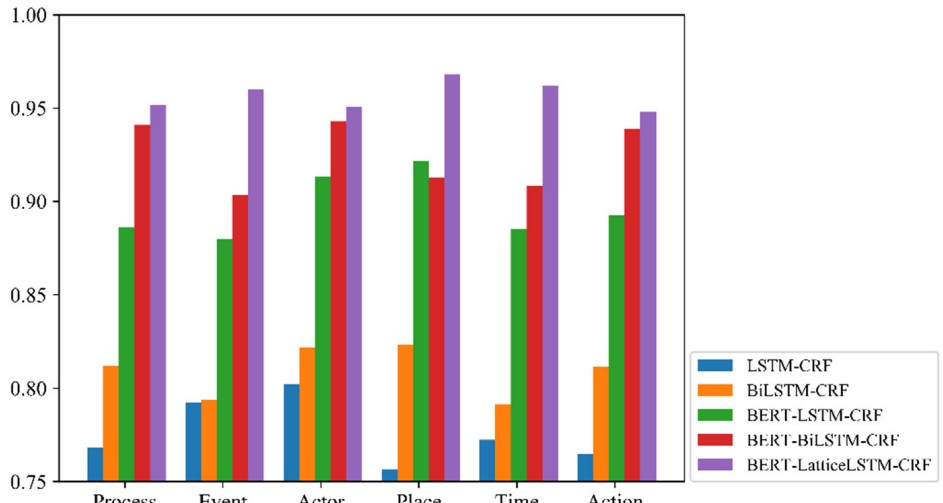

**Figure 7.** Test results for the named entity recognition performance for different entity types (%).

Figure 8 shows the variation of the F1-measure with respect to the number of iterations for the proposed, LSTM-CRF, BiLSTM-CRF, BERT-LSTM-CRF, and BERT-BiLSTM-CRF models. As the number of iterations increased, the F1-measure of the proposed model increased and stabilized. Moreover, compared with the other models, the proposed model achieved a higher F1-measure throughout the iteration process, confirming its superior for improving the named entity recognition performance.

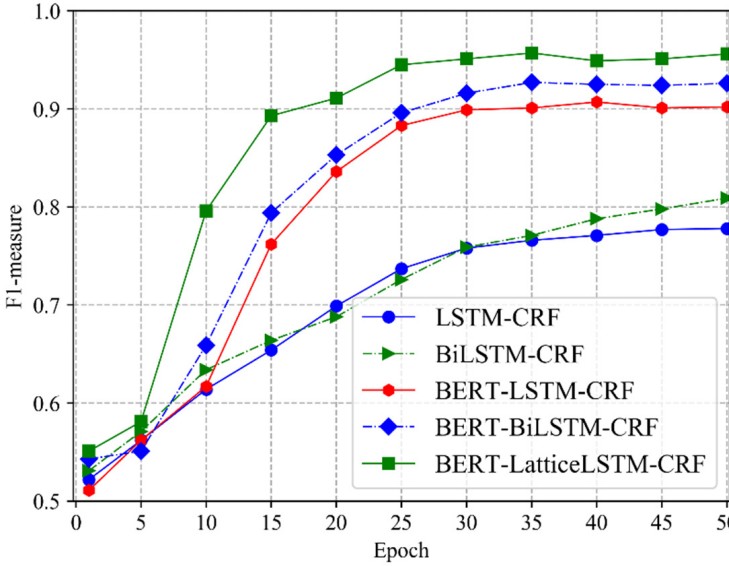

**Figure 8.** Curves of the F1-measure with respect to the number of iterations.

### 3.4. Performance Validation of Proposed Model for Relation Extraction

We experimentally evaluated the performance of the proposed relation extraction model in comparison with the LSTM-RC, BiLSTM-RC, BERT-LSTM-RC, and BERT-BiLSTM-RC models. Here, RC represents the relation classifier proposed in Section 3.3. The test results for relation extraction performance are presented in Table 4.

**Table 4.** Test results for the relation extraction performance (%).

| Model | Precision | Recall | F1-Measure |
|---|---|---|---|
| LSTM-RC | 72.55 | 74.69 | 73.60 |
| BiLSTM-RC | 78.24 | 79.16 | 78.70 |
| BERT-LSTM-RC | 86.86 | 83.64 | 85.22 |
| BERT-BiLSTM-RC | 91.22 | 92.37 | 91.79 |
| BERT-Lattice-LSTM-RC | **95.89** | **96.03** | **95.96** |

As shown in Table 4, the proposed relation extraction model achieved the highest Precision, Recall, and F1-measure values, indicating its superior performance. Compared with the LSTM-RC and BiLSTM-RC models, the BERT-LSTM-RC model exhibited improvements of 14.31% and 8.62% in Precision, 8.95% and 3.88% in Recall, and 11.62% and 6.52% in F1-measure, respectively. The BERT-BiLSTM-RC model exhibited improvements of 18.67% and 12.98% in Precision, 17.68% and 13.21% in Recall, and 18.19% and 13.09% in F1-measure, respectively. Furthermore, the BERT-Lattice-LSTM-RC model exhibited improvements of 23.34% and 17.65% in Precision, 21.34% and 16.87% in Recall, and 22.36% and 17.26% in F1-measure, respectively. This indicates that utilizing BERT-based pretrained models can enhance the named entity recognition performance. Moreover, compared with the BERT-LSTM-RC and BERT-BiLSTM-RC models, the BERT-Lattice-LSTM-RC model exhibited improvements of 9.03% and 4.67% in Precision, 12.39% and 3.66% in Recall, and 10.74% and 4.17% in F1-measure, respectively. This suggests that incorporating the Lattice-LSTM model can integrate implicit lexical features into the character-based LSTM model, thereby improving the relation extraction performance. The experimental results confirm the effectiveness of the BERT model and the Lattice-LSTM model for enhancing the relation extraction performance.

The data of Table 4 are displayed as a bar chart in Figure 9. As shown, the proposed model outperformed the LSTM-RC, BiLSTM-RC, BERT-LSTM-RC, and BERT-BiLSTM-RC models for all three evaluation metrics: Precision, Recall, and F1-measure. This confirms the superiority of the proposed model for relation extraction.

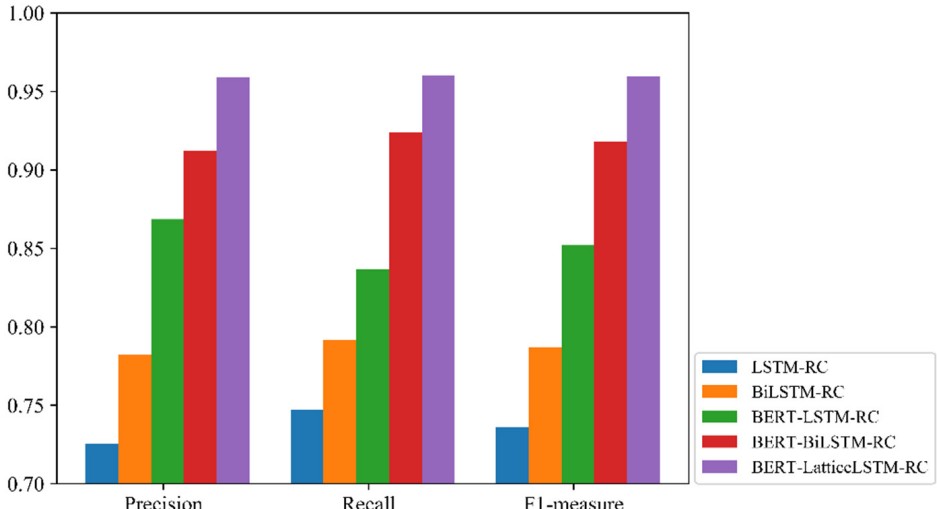

**Figure 9.** Test results for the relation extraction performance (%).

Furthermore, we conducted tests on seven relation types—hasEvent, hasActor, has-Place, hasTime, hasAction, cause, and followed—to evaluate the F1-measures of the LSTM-RC, BiLSTM-RC, BERT-LSTM-RC, BERT-BiLSTM-RC, and BERT-Lattice-LSTM-RC models in recognizing different relation types. The results are presented in Table 5.

**Table 5.** Test results for the relation extraction performance for different relation types.

| Model | has-Event | has-Actor | has-Place | hasTime | has-Action | Cause | Followed |
|---|---|---|---|---|---|---|---|
| LSTM-RC | 72.05 | 72.88 | 73.41 | 71.61 | 73.87 | 75.23 | 76.15 |
| BiLSTM-RC | 76.91 | 75.27 | 78.16 | 79.22 | 76.92 | 81.14 | 83.28 |
| BERT-LSTM-RC | 86.20 | 85.89 | 82.69 | 84.37 | 85.10 | 86.13 | 86.16 |
| BERT-BiLSTM-RC | 90.21 | 91.38 | 91.37 | 89.72 | 90.84 | 95.30 | 93.71 |
| BERT-Lattice-LSTM-RC | **94.18** | **94.55** | **95.31** | **95.77** | **96.01** | **98.13** | **97.77** |

As shown in Table 5, compared with the other models, the proposed model achieved F1-measure improvements ranging from 3.97% to 22.13% for the hasEvent relation, from 3.17% to 21.67% for the hasActor relation, from 3.94% to 21.90% for the hasPlace relation, from 6.05% to 24.16% for the hasTime relation, from 5.17% to 22.14% for the hasAction relation, from 2.83% to 22.90% for the cause relation, and from 4.06% to 21.62% for the followed relation. The proposed model achieved the highest F1-measure values for different relation types, further highlighting its superiority for relation extraction.

The data in Table 5 are displayed as a bar chart in Figure 10. As shown, the proposed model had a significantly higher F1-measure than the LSTM-RC, BiLSTM-RC, BERT-LSTM-RC, and BERT-BiLSTM-RC models for all seven entity types: hasEvent, hasActor, hasPlace, hasTime, hasAction, cause, and followed. This confirms the superiority of the proposed model for relation extraction.

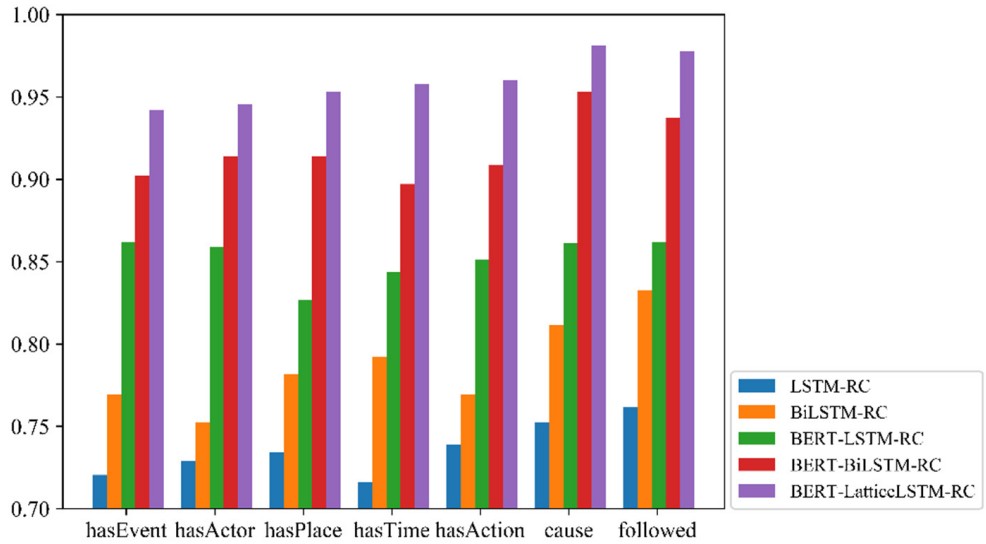

**Figure 10.** Test results for the relation extraction performance for different relation types (%).

Figure 11 presents the variation of F1-measure of the proposed, LSTM-RC, BiLSTM-RC, BERT-LSTM-RC, and BERT-BiLSTM-RC models with respect to the number of iterations. As shown, as the number of iterations increased, the F1-measure of the proposed model improved and reached a stable state. Moreover, compared with the LSTM-RC, BiLSTM-RC, BERT-LSTM-RC, and BERT-BiLSTM-RC models, the proposed model achieved a higher F1-measure throughout the iteration process, confirming its effectiveness for improving the relation extraction performance.

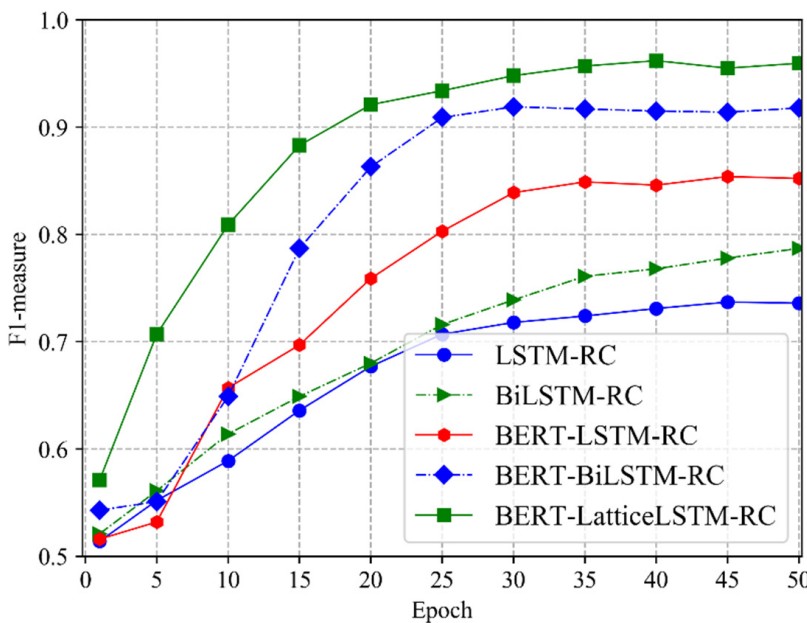

**Figure 11.** Curves of the F1-measure of the models for relation extraction with respect to the number of epochs.

*3.5. Performance Validation of Proposed Model for Triplet Extraction*

The triplet extraction performance of the proposed model was experimentally evaluated in comparison with the LSTM-CRF-RC, BiLSTM-CRF-RC, BERT-LSTM-CRF-RC, and BERT-BiLSTM-CRF-RC models. The triplet extraction in the experiment involved two steps. First, named entities were extracted from Chinese sentences using named entity recognition models (LSTM-CRF, BiLSTM-CRF, BERT-LSTM-CRF, BERT-BiLSTM-CRF, BERT-Lattice-LSTM-CRF). Then, the Chinese sentences were labeled and fed into the relation extraction model (LSTM-RC, BiLSTM-RC, BERT-LSTM-RC, BERT-BiLSTM-RC, BERT-Lattice-LSTM-RC), which output the relation between the two labeled named entities, resulting in the final triplet. The test results for the triplet extraction performance are presented in Table 6.

**Table 6.** Test results for the triplet extraction performance (%).

| Model | Precision | Recall | F1-Measure |
|---|---|---|---|
| LSTM-CRF-RC | 56.70 | 57.56 | 57.13 |
| BiLSTM-CRF-RC | 64.27 | 63.06 | 63.66 |
| BERT-LSTM-CRF-RC | 78.76 | 74.14 | 76.38 |
| BERT-BiLSTM-CRF-RC | 85.18 | 84.56 | 84.87 |
| BERT-Lattice-LSTM-CRF-RC | **92.09** | **91.51** | **91.80** |

As shown in Table 6, the proposed model achieved the highest Precision, Recall, and F1-measure values, indicating its superior triplet extraction performance. Compared with the LSTM-CRF-RC and BiLSTM-CRF-RC models, the BERT-LSTM-CRF-RC model exhibited improvements of 22.06% and 14.49% in Precision, 16.58% and 11.08% in Recall, and 19.25% and 12.72% in F1-measure, respectively. The BERT-BiLSTM-CRF-RC model exhibited improvements of 28.48% and 20.91% in Precision, 27.00% and 21.50% in Recall, and 27.74% and 21.21% in F1-measure, respectively. Additionally, the BERT-Lattice-LSTM-RC model achieved improvements of 35.39% and 27.82% in Precision, 33.95% and 28.45% in Recall, and 34.67% and 28.14% in F1-measure, respectively. These results indicate that using BERT-based pretraining models can enhance the triplet extraction performance. Furthermore, compared with the BERT-LSTM-CRF-RC and BERT-BiLSTM-CRF-RC models, the BERT-Lattice-LSTM-RC model exhibited improvements of 13.33% and 6.91% in Precision,

17.37% and 6.95% in Recall, and 15.42% and 6.93% in F1-measure. This suggests that incorporating the Lattice-LSTM model can fuse implicit lexical features into the character-based LSTM model, enhancing the relation extraction performance. The experimental results confirm the effectiveness of the BERT model and Lattice-LSTM model for improving the triplet extraction performance.

Furthermore, compared with the test results presented in Tables 2 and 4, the triplet extraction performance of all the models exhibited reductions of varying degrees in terms of the Precision, Recall, and F1-measure. This was due to the introduction of errors in the named entity recognition during the relation extraction process, which resulted in error accumulation and degraded the triplet extraction performance. However, with regard to the triplet extraction performance, the proposed model exhibited a higher degree of improvement in the Precision, Recall, and F1-measure than the other models. This confirms the superior named entity recognition and relation extraction performance of the proposed model, which led to superior triplet extraction performance.

The data in Table 6 are displayed as a bar chart in Figure 12. As shown, the proposed model outperformed the LSTM-CRF-RC, BiLSTM-CRF-RC, BERT-LSTM-CRF-RC, and BERT-BiLSTM-CRF-RC models for all three evaluation metrics: Precision, Recall, and F1-measure. This confirms the superiority of the proposed model for triplet extraction.

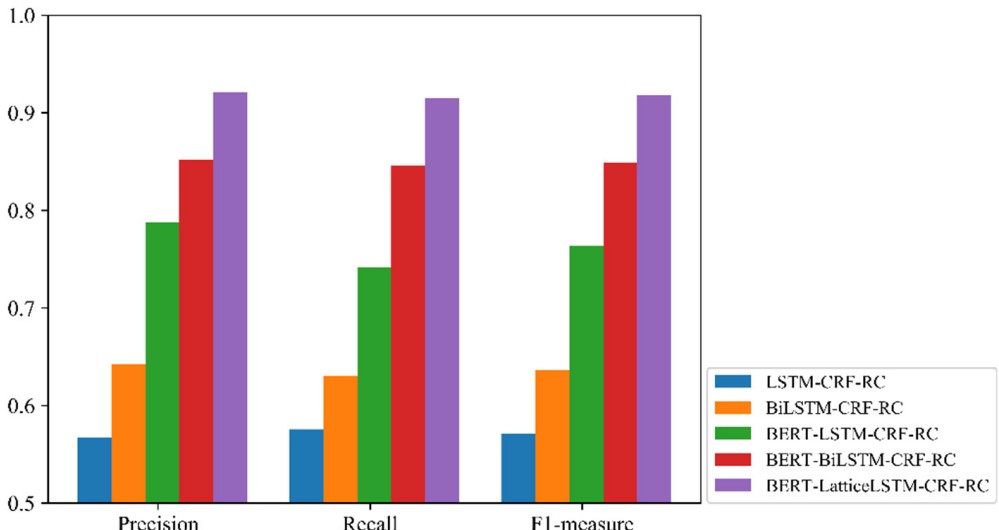

**Figure 12.** Test results for the triplet extraction performance (%).

## 4. Conclusions

This paper presents a method for constructing a spatiotemporal knowledge graph focusing on maritime ship activities, which includes the design of an ontology layer and a population method for the data layer. The ontology layer includes an SAM that describes conceptual entities and their relations in ship activities, consisting of six core entity concepts (process, event, actor, place, time, action) and seven relation types (hasEvent, hasActor, hasPlace, hasAction, cause, followed). A ship activity entity–relation triplet extraction model based on the BERT-Lattice-LSTM-CRF-RC model is proposed for populating the data layer of the knowledge graph. First, the text statements are fed into the BERT model for pretraining to obtain character-level vector representations. These representations are then input into the Lattice-LSTM model for processing, and the resulting hidden vectors $h_1, h_2, \cdots, h_n$ are passed through a CRF model for named entity recognition. The recognized named entities are marked in the original text statements and then fed into another BERT-Lattice-LSTM network model. The hidden vectors $h'_1, h'_2, \cdots, h'_n$ generated by this model are input into an RC, and the output represents the relation between the two marked named entities. This completes the extraction of entity–relation triplets. In experiments, the proposed method achieved triplet extraction performance exceeding 90% for three different evaluation metrics: Precision, Recall, and F1-measure. This confirmed that

the proposed model is effective for the construction of spatiotemporal knowledge graphs for maritime ship activities.

The limitations of this study lie in the lack of making full use of ship's activity track data to carry out activity behavior analysis. In the future work, we will further optimize the method and study the trajectory semantic method to convert ship trajectory data into semantic information, so as to complete the construction and application of spatio-temporal knowledge graph.

**Author Contributions:** Conceptualization, C.X.; methodology, C.X.; software, C.X.; validation, C.X., L.Z. and Z.Z.; writing—original draft preparation, C.X.; writing—review and editing, C.X., L.Z. and Z.Z.; funding acquisition, L.Z. All authors have read and agreed to the published version of the manuscript.

**Funding:** This research was supported in part by the National Natural Science Foundation of China under Grant 91538201, in part by Taishan Scholar Project of Shandong Province under Grant ts201511020, and in part by Project supported by Chinese National Key Laboratory of Science and Technology on Information System Security under Grant 6142111190404.

**Data Availability Statement:** The data supporting the reported results are available from the corresponding author upon reasonable request.

**Conflicts of Interest:** The authors declare no conflict of interest.

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
