# Peer review of "A Novel Method for Constructing Spatiotemporal Knowledge Graph for Maritime Ship Activities"

_electronics, doi:10.3390/electronics12153205_

Round 1

Reviewer 1 Report

The article is devoted to the development of an ontological approach to the practical task of describing the activity of sea vessels.

In my opinion, the main shortcomings are poor structuring and overloading of the paper with commonly well-known material. Specific comments are as follows.

1. The introduction is too long. It should be significantly shortened, taking away well-known material and moving part of the Introduction to the next chapter 2, focusing on detailed analysis and comparison with the results of similar research results.

2. Sections 2 and 3 should be combined into one - research methods and materials.

3. In Chapter 3, some of the formulas are common knowledge, for example, formula 25. Similarly, in Chapter 4, formulas 26-28 are also common knowledge. It is generally not recommended to submit publicly known material in this section. These formulas should be removed, only to mention them it would be enough (give a link). Note that the authors use them and indicate why. Otherwise, Chapter 3 seems to be part of a lecture for students.

4. Chapter 4 is experimental. Although there are interesting results, I recommend splitting this section into two:

- description of the experiment,

- obtained results.

In addition, I recommend authors to add a new section "Discussion of results and their analysis". In this way, the research material will be better perceived by the readers. Conclusion: the article is interesting, the topic is relevant, the results are important from a practical point of view, but the main observation is that the paper requires significant restructuring and removal of well-known material.

The level of English is sufficient.

Author Response

Reviewer#1, Concern # 1: The introduction is too long. It should be significantly shortened, taking away well-known material and moving part of the Introduction to the next chapter 2, focusing on detailed analysis and comparison with the results of similar research results.

Author response: 

Thank you for your suggestion. We have taken away well-known material and moved part of the Introduction to the next chapter 2

Reviewer#1, Concern # 2: Sections 2 and 3 should be combined into one - research methods and materials.

Author response: 

Thank you for your suggestion. We have combined section 2 and section 3 into one - research methods and materials.

Reviewer#1, Concern # 3: In Chapter 3, some of the formulas are common knowledge, for example, formula 25. Similarly, in Chapter 4, formulas 26-28 are also common knowledge. It is generally not recommended to submit publicly known material in this section. These formulas should be removed, only to mention them it would be enough (give a link). Note that the authors use them and indicate why. Otherwise, Chapter 3 seems to be part of a lecture for students.

Author response: 

Althouth some of the formulas are common knowledge, but they are the basis of the new knowledge, that is, the new methods proposed, and it is necessary to present them in the paper for readers to understand. Therefore, they cannot be deleted.

Reviewer#1, Concern # 4: Chapter 4 is experimental. Although there are interesting results, I recommend splitting this section into two:

- description of the experiment,

- obtained results.

In addition, I recommend authors to add a new section "Discussion of results and their analysis". In this way, the research material will be better perceived by the readers. Conclusion: the article is interesting, the topic is relevant, the results are important from a practical point of view, but the main observation is that the paper requires significant restructuring and removal of well-known material.

Author response: 

In fact, description of the experiment, obtained results, and discussion of results and their analysis, all three parts are reflected in the part of Experiment, and the three parts fit together organically.

Reviewer 2 Report

  • Overall, the paper is well-written and technically sound. However, the minor comments above could be addressed to make the paper more readable and accessible to a wider audience.
  • The paper does not provide a clear definition of what a "ship activity" is. This makes it difficult to understand the scope of the work and the relevance of the results.
  • The paper does not provide a comprehensive evaluation of the proposed method. The only evaluation metric used is F1-score, which is a measure of accuracy. Other metrics, such as precision and recall, should also be used to provide a more complete picture of the performance of the method.
  • The paper does not discuss the limitations of the proposed method. For example, the method is only evaluated on a small dataset of maritime ship activities. It is not clear how the method would perform on a larger dataset or on a different domain.
  • The paper uses a very complex and technical approach to the problem of constructing spatiotemporal knowledge graphs. This makes it difficult for readers to understand the method and its implications.
  • The paper does not provide enough detail about the implementation of the proposed method. This makes it difficult to reproduce the results or to extend the method to other applications.
  • The paper does not address the problem of data sparsity. This is a major challenge in the construction of spatiotemporal knowledge graphs, and the proposed method does not seem to address it effectively.

Author Response

Reviewer#2, Concern # 1: The paper does not provide a comprehensive evaluation of the proposed method. The only evaluation metric used is F1-score, which is a measure of accuracy. Other metrics, such as precision and recall, should also be used to provide a more complete picture of the performance of the method.

Author response: 

F1-score is a comprehensive evaluation metric, which covers the characteristics of precision and recall.  Therefore, we believe that using F1-score as an evaluation metric can be a good method to evaluate the performance of the proposed method

Reviewer#2, Concern # 2: The paper does not discuss the limitations of the proposed method. For example, the method is only evaluated on a small dataset of maritime ship activities. It is not clear how the method would perform on a larger dataset or on a different domain.

Author response: 

Thank you for your advice. Our proposed method uses knowledge graph for the first time to analyze ship activity behavior. We will further study your suggestions and questions as future work

Reviewer#2, Concern # 3: The paper uses a very complex and technical approach to the problem of constructing spatiotemporal knowledge graphs. This makes it difficult for readers to understand the method and its implications.

Author response: 

Because of the complexity of Chinese text data, it is necessary to match the named entity recognition and relation extraction methods, which leads to the complexity of the algorithm

Reviewer#2, Concern # 4: The paper does not provide enough detail about the implementation of the proposed method. This makes it difficult to reproduce the results or to extend the method to other applications.

Author response: 

Due to length constraints, the paper cannot provide sufficient detail on the implementation of the proposed method.

Reviewer#2, Concern # 5: The paper does not address the problem of data sparsity. This is a major challenge in the construction of spatiotemporal knowledge graphs, and the proposed method does not seem to address it effectively.

Author response: 

Thank you for your advice. Our proposed method uses knowledge graph for the first time to analyze ship activity behavior. We will further study your suggestions and questions as future work.

Reviewer 3 Report

This manuscript depicts the setting of a spatiotemporal ship activity dataset.

The result of this endeavour is useful and topical. Moreover, I cannot spot any significant technical weakness.

However, the main issue is that this work is exactly the construction of the knowledge graph, as the title explains. The fact that it includes a series of procedures, does not make it a novel method, especially because there is no breakthrough in any of the steps. Those are known techniques, now combined to produce this database.

The redaction is not proficient, but there are not significant errors.

Author Response

Reviewer#3, Concern # 1: However, the main issue is that this work is exactly the construction of the knowledge graph, as the title explains. The fact that it includes a series of procedures, does not make it a novel method, especially because there is no breakthrough in any of the steps. Those are known techniques, now combined to produce this database.

Author response: 

The main contribution of our paper is to solve the problem of ship activity analysis by using knowledge graph as a research tool for the first time in view of the complexity of ship activity analysis

Reviewer 4 Report

​The paper presents a method for constructing a spatiotemporal knowledge graph for maritime ship activities, addressing an important challenge in the field. The authors propose a ship activity ontology model to define the entities and relationships, which serves as a foundation for constructing the knowledge graph.

The utilization of maritime event text data as the ship activity dataset is a practical choice, as it reflects real-world scenarios and enhances the applicability of the method. The extraction of entities and relations from the text data to form triplets showcases a systematic approach to knowledge graph construction. The use of pretraining with the BERT model to obtain vector representations of characters is a well-founded strategy given BERT's proven performance in natural language processing tasks.

The introduction of the lattice long short-term memory network (Lattice-LSTM) for further processing of the vector representations is a noteworthy addition. This approach has the potential to capture long-range dependencies and contextual information, which can improve the accuracy of subsequent steps. The incorporation of a conditional random field (CRF) for named entity recognition is a suitable choice as it leverages the dependencies among the labeled entities, enhancing the quality of the extracted triplets.

The experimental results presented in the paper demonstrate the effectiveness of the method. Achieving triplet extraction performance exceeding 90% for precision, recall, and F1-measure metrics highlights the robustness of the approach. 

  However, it would have been beneficial if the authors had provided a comparison with existing methods or alternative approaches to further highlight the advantages of their proposed method.   ​Also, the presentation of the paper is weak ​including the figures and the mathematical formulae. 

minor issues 

Author Response

Reviewer#4, Concern # 1:   However, it would have been beneficial if the authors had provided a comparison with existing methods or alternative approaches to further highlight the advantages of their proposed method.  ​Also, the presentation of the paper is weak ​including the figures and the mathematical formulae.

Author response: 

Unfortunately, there are few relevant studies on ship activities, so relevant methods are scarce and it is difficult to carry out comparative experiments.

Reviewer 5 Report

The manuscript is well written and organized. The argument is original and aligned with the scope of the journal. According to my opinion it can be accepted for publication after minor improvements.

ABSTRACT

Please ensure that the abstract has the following elements: 1-2 sentences on the context and the need for the study; 1-2 sentences on the methodology; the majority of the abstract on the actual results of the study; 1-2 sentences on key conclusions and recommendations.

INTRODUCTION

In the introduction, the novelty of the contribution must be better specified in relation to the previous contributions in literature.

PROPOSED METHOD AND DISCUSSION

The Chinese writing in figures 3 and 4 should be translated directly into the images.

DISCUSSION OF THE RESULTS

The method presented is very interesting, as well as the way to use the BERT. I ask the authors how the extraction of numerical values (as well as textual ones) from sentences can be treated with this method? I am convinced that it can do it but I would like to see it written explicitly, so as to enhance it more. Furthermore, I would suggest writing something on the integration of the proposed method with other methods to show its validity on a more general level, even beyond the case study. For example, to extract numeric values for other purposes. For example, consider the integration (among future developments) with the framework of: “Spreafico, C., Landi, D., & Russo, D. (2023). A new method of patent analysis to support prospective life cycle assessment of eco-design solutions. Sustainable Production and Consumption, 38, 241-2512”.

CONSLUSIONS

In the conclusions, the limitations of this study should be better specified.

Author Response

Reviewer#5, Concern # 1:   Please ensure that the abstract has the following elements: 1-2 sentences on the context and the need for the study; 1-2 sentences on the methodology; the majority of the abstract on the actual results of the study; 1-2 sentences on key conclusions and recommendations.

Author response: 

We have ensured that the abstracts include the elements mentioned by the reviewer.

Reviewer#5, Concern # 2: The Chinese writing in figures 3 and 4 should be translated directly into the images.

Author response: 

Thank you for your suggestion. We have made relevant modifications according to the suggestion of reviewer

Reviewer#5, Concern # 3: The method presented is very interesting, as well as the way to use the BERT. I ask the authors how the extraction of numerical values (as well as textual ones) from sentences can be treated with this method? I am convinced that it can do it but I would like to see it written explicitly, so as to enhance it more. Furthermore, I would suggest writing something on the integration of the proposed method with other methods to show its validity on a more general level, even beyond the case study. For example, to extract numeric values for other purposes. For example, consider the integration (among future developments) with the framework of: “Spreafico, C., Landi, D., & Russo, D. (2023). A new method of patent analysis to support prospective life cycle assessment of eco-design solutions. Sustainable Production and Consumption, 38, 241-2512”.

Author response: 

Due to length constraints, the paper cannot provide sufficient detail on the implementation of the proposed method.

Reviewer#5, Concern # 4: In the conclusions, the limitations of this study should be better specified.

Author response: 

We have presented the limitations of this study in the conclusions, which are as followed:

The limitations of this study lie in the lack of making full use of ship's activity track data to carry out activity behavior analysis. In the future work, we will further optimize the method and study the trajectory semantic method to convert ship trajectory data into semantic information, so as to complete the construction and application of spatio-temporal knowledge graph

Round 2

Reviewer 4 Report

The paper can be accepted.

English is fine